# Effect of Early Management on Pain and Depression in Patients with Pancreatobiliary Cancer: A Randomized Clinical Trial

**DOI:** 10.3390/cancers11010079

**Published:** 2019-01-11

**Authors:** Sang Myung Woo, Mi Kyung Song, Meeyoung Lee, Jungnam Joo, Dae Hyun Kim, Jong-Heun Kim, Sung-Sik Han, Sang-Jae Park, Tae Hyun Kim, Woo Jin Lee

**Affiliations:** 1Pancreatobiliary Cancer Clinic, National Cancer Center, Goyang 10408, Korea; leemyyoung@ncc.re.kr (M.L.); sshan@ncc.re.kr (S.-S.H.); spark@ncc.re.kr (S.-J.P.); k2onco@ncc.re.kr (T.H.K.); 2Biometrics Research Branch and Biostatistics Collaboration Unit, National Cancer Center, Goyang 10408, Korea; smk@ncc.re.kr or songmk@nhis.or.kr (M.K.S.); jooj@ncc.re.kr (J.J.); 3Health Insurance Policy Research Institute, National Health Insurance Service, 199, Hyeoksin-ro, Wonju 26465, Korea; 4Department of Anesthesiology and Pain Medicine, National Cancer Center, Goyang 10408, Korea; dhkim@ncc.re.kr; 5Mental Health Clinic, National Cancer Center, 10408 Goyang, Korea; psy@ncc.re.kr

**Keywords:** pancreatic cancer, biliary tract cancer, pain, depression, palliative care

## Abstract

*Background:* The present study assessed whether early palliative care (EPC) targeting pain and depression and automated symptom monitoring could improve symptoms in patients with advanced pancreatobiliary cancer. *Methods:* Patients diagnosed with pathologically confirmed locally advanced or metastatic pancreatic or biliary tract cancer who had cancer-related pain (brief pain inventory (BPI) worst pain score >3) and/or depression (Center for Epidemiological Studies—Depression Scale (CES-D) >16) were randomized within 8 weeks after diagnosis to receive EPC or on-demand palliative care (*n* = 144 each). EPC included (1) nursing assessment of pain and depression, (2) pain control based on National Comprehensive Cancer Network guidelines, (3) depression control by psychoeducation and/or consultation with a psychiatric specialist, and (4) patient education. The primary end points were ≥50% reductions from baseline to week 4 in pain and depression scores. *Results*: The proportion of patients in the EPC and usual care groups with ≥50% reductions in pain (29.5% vs. 25.2%; *p* = 0.4194) and depression (30.8% vs. 36.8%; *p* = 0.5732) scores from baseline to week 4 did not differ significantly. The proportion of patients with BPI worst pain score ≤3 was significantly higher (51.1% vs. 38.9%, *p* = 0.0404) and the reduction in pain intensity score significantly greater (1.5 vs. 1.0 points, *p* = 0.0318) in the EPC than in the usual care group. At 4 weeks, patients in the EPC group reported significant increases in global health status, role of functioning, nausea and vomiting, and pain scores on the European Organization for Research and Treatment of Cancer Core Quality of Life questionnaire (EORTC QLQ-C30) general questionnaire. *Conclusions:* Although the primary outcome was not met, this trial indicates that EPC may improve early pain relief in patients with advanced pancreatobiliary cancers.

## 1. Introduction

Pancreatic cancer (PC) and biliary tract cancer (BTC) are frequently diagnosed at a late stage, preventing resection. Moreover, tumors may progress despite anticancer treatment. Appropriate treatment plans that improve survival while maintaining subjective quality of life (QoL) are essential. Because the pancreas is anatomically located in the central abdomen at the root of the mesentery and because BTC metastasizes primarily to lymph nodes in the hepatoduodenal ligament and to para-aortic nodes, most patients with PC and BTC have a significant symptom burden and require frequent medical attention and hospitalization for symptom management [1,2]. The marked reduction in physical efficiency associated with advanced cancers and their induction of persistent chronic pain interfere with the physical and psychological integrity of patients, indicating a need for both symptom control and disease-modifying therapy [3]. Palliative care is therefore important in patients with pancreatobiliary cancer. 

Early referral of lung cancer patients to palliative care in addition to standard oncology care was found to improve survival and QoL compared with standard oncology care alone [4]. However, a recent randomized clinical trial showed that, although early integrated palliative care improved QoL and other prominent outcomes, these effects differed by type of cancer [5]. These interventions had significant effects in patients with lung cancer but not in patients with gastrointestinal cancers, including pancreatobiliary cancer. 

Symptoms frequently observed in patients with PC or BTC include pain, depression and anxiety, anorexia, cachexia, weight loss, pancreatic exocrine insufficiency, jaundice, and gastric outlet obstruction. However, despite the availability of effective analgesics and new technologies for drug administration, pain management remains suboptimal [6]. Pain affects approximately 80% of patients with PC [7,8] and 50% of patients with BTC [9]. Adequate control of pain is often not achieved due to limited treatment options and significant variations in local practice, emphasizing the need for new approaches [6]. In addition, the risk of depression is significantly higher in cancer patients than in the general population, with the highest prevalence of depression observed in patients with pancreatic tumors [10]. In Korea, suicide rates are especially high among patients with PC and BTC [11]. 

Many palliative care specialists question the feasibility, practicability, and efficiency of this approach. Specialist palliative care has shown minimal effects on QoL [12], with the reported effect sizes smaller than expected. In addition, resource allocation issues interfere with the ability to refer all advanced cancer patients at the palliative stage of their diseases to specialist teams [13,14]. The aim of the present study was to determine whether a selective approach to early palliative care (EPC), targeting pain and depression, integrated with usual oncologic care with automated symptom screening and monitoring, can improve pain and depression in patients with advanced pancreatobiliary cancer.

## 2. Methods

### 2.1. Study Design

Patients diagnosed with pathologically confirmed locally advanced or metastatic PC or BTC at the National Cancer Center, Korea, between April 2012 and May 2016 and who had cancer-related pain (brief pain inventory (BPI) worst pain score >3) and/or depression (Center for Epidemiological Studies—Depression Scale (CES-D) >16) were randomized 1:1 within 8 weeks after diagnosis to receive either EPC or usual oncology care (*n* = 144 each). Patients were stratified based on tumor type (PC or BTC) and symptom type (pain only, depression only, and pain and depression).

The primary endpoints were 50% reductions from baseline to week 4 in pain score and depression score, as determined by the BPI worst pain score and the CES-D, respectively. Secondary endpoints included improvements in QoL, quantitative pain relief, sleep disturbance, satisfaction with pain control, patient’s and investigator’s global assessment and CGI-I (clinical global impression–improvement) scores, and overall survival. All the participants continued to receive routine oncologic care throughout the study period. The protocol was approved by the institutional review board of National Cancer Center, Korea (IRB No., NCC-CTC-12-605). All participants provided written informed consent.

### 2.2. Patients

Patients who presented to the outpatient cancer clinic were invited by their physicians to enroll in the study; all the physicians in the clinic agreed to approach, recruit, and obtain consent from their patients. Physicians were encouraged, but not required, to offer participation to all eligible patients; no additional screening or recruitment measures were used. Patients were eligible to participate if they had pathologically confirmed locally advanced or metastatic PC or BTC diagnosed within the previous 8 weeks, a Karnofsky performance rating scale ≥50% and cancer-related pain (BPI worst pain score >3) and/or depression (CES-D >16).

### 2.3. Intervention

EPC included: (1) Nursing assessment of pain and depression, (2) pain control based on National Comprehensive Cancer Network (NCCN) guidelines, (3) depression control by psychoeducation and/or consultation with a psychiatric specialist, and (4) patient education. Nursing assessment of pain included a brief evaluation of each patient’s mood state with the CES-D. Patients were managed by three research nurses, led by Meeyoung Lee, trained in assessing symptom response and medication adherence; in providing specific education about pain and depression; and in making treatment adjustments according to NCCN guidelines. Patients with CED-S scores >25 were referred to psychiatric specialists. We delivered the interventions by telephone or during regularly scheduled outpatient care. Follow-up intervention visits or telephone coaching were scheduled daily until BPI worst pain score was ≤3. Participants were contacted at baseline and at 1, 3, 6, 9, and 12 months. In addition, telephone calls were triggered when patients reported inadequate symptom improvement, nonadherence to medication, adverse effects, or suicidal ideation, or when patients requested to be contacted.

### 2.4. Usual Oncology Care

Patients randomized to the usual-care group received no formal intervention but were informed of their depressive and pain symptoms; moreover, their screening results were provided to their physician. Usual oncology care (UOC), provided to all patients, was directed by an attending physician and consisted of anticancer and symptom control treatments and consultation with psychiatric and pain care specialists. The latter were provided whenever requested, regardless of group assignment.

### 2.5. Patient-Reported Measures

Health-related QoL was measured using the European Organization for Research and Treatment of Cancer Core Quality of Life questionnaire (EORTC QLQ-C30) general questionnaire, Korean version, which assesses multiple dimensions of QoL (physical, functional, emotional, and social well-being) during the previous week. Patients were also assessed using the Insomnia Severity Index (ISI), a 7-item self-reported questionnaire assessing the nature, severity, and impact of insomnia [15]. Each item was rated on a 5-point Likert scale (0 = no problem; 4 = very severe problem), yielding a total score ranging from 0 to 28. 

Satisfaction with pain management was measured on a 5-point scale (1 = very good; 5 = very poor) [16]. The 5-point patient’s and investigator’s global assessments were used as outcome measures with a score of 5 defined as very effective [17]. Clinicians evaluated patients on the CGI-I scale, a 7-point scale measuring the improvement or worsening of a patient relative to baseline, with 1 = very much improved; 7 = very much worse [18].

Mood was assessed by caregivers using the CES-D, a 20-item questionnaire that rates the frequency during the previous week of symptoms associated with depression, including restless sleep, poor appetite, and feeling lonely. Each item was rated on a scale from 0 to 3, with total scores ranging from 0 to 60. Higher scores indicated greater depressive symptoms, with scores ≥16 indicating individuals at risk for clinical depression, with good sensitivity and specificity and high internal consistency [19].

### 2.6. Statistical Analyses

The co-primary endpoints were defined as 50% reductions from baseline in BPI pain score and CES-D score after 4 weeks. The target sample size, calculated for a power of 80% and a corrected two-sided alpha of 0.025 and allowing a drop-out rate of 18–20%, was 144 per group. Analyses were based on intention to-treat (ITT) in all randomized participants. To assess sensitivity, the primary endpoints were also analyzed for a modified ITT set, which excluded subjects who did not meet the critical inclusion criteria in the ITT set. The missing value was imputed to the negative results because it was considered that the cause of the missing value was not effective for treatment. All other analyses for secondary purposes excluded missing data.

Continuous variables were expressed as median and range and categorical variables as frequency (%). Between-group differences in continuous variables were compared using Mann–Whitney U-tests, and between-group differences in categorical variables were compared using the Chi-square or Fisher’s exact test, as appropriate. Variables at baseline and after four weeks within each group were compared using Wilcoxon signed-rank tests. Survival rates were assessed by the Kaplan–Meier method and compared using a log-rank test. All statistical analyses were two-sided, and statistical significance was defined as *p* < 0.05, except for the primary endpoint. All analyses were performed using Statistical Analysis System (version 9.3, SAS Institute, Inc., Cary, NC, USA) and R software (version 3.3.3; R Foundation for Statistical Computing, Vienna, Austria).

## 3. Results

### 3.1. Participant Enrollment and Baseline Characteristics

Figure 1 summarizes the participant flow in this trial. Of the 376 patients screened, 309 were positive for pain, depression, or both, and 313 consented to enroll in the study. Of these, 144 patients respectively were randomized to the intervention and usual-care groups. By April 2017, 243 (84.4%) patients had died. Among participants alive at each follow-up point, assessment rates were similar in both groups and uniformly high, with 93.4% (228 of 244) participation at 1 month, 86.2% (150 of 174) at 3 months, 83.3% (90 of 108) at 6 months, 83.8% (62 of 74) at 9 months, and 72.3% (34 of 47) at 12 months.

Baseline characteristics were balanced in the two groups (Table 1, Appendix A). Of the 288 patients, 211 (73.3%) had pain only, 10 (3.5%) had depression only, and 67 (23.3%) had both. The median ages of patients in the intervention and control groups were 66.00 and 67.00 years, respectively, and 44.44% and 45.14%, respectively, were men. At baseline, the two groups showed no difference in scores for measures of QoL (*p* = 0.8035) and depressive symptoms (*p* = 0.2089).

### 3.2. Pain and Depression-specific Outcomes

There was no significant difference in the proportion of patients in the EPC and usual care groups with ≥50% reductions in pain (29.5% vs. 25.2%; *p* = 0.4194) and depression (30.8% vs. 36.8%; *p* = 0.5732) scores from baseline to week 4 (Table 2). The proportion of patients with BPI worst pain score ≤3 was significantly higher in the EPC than in the usual care group. (51.1% vs. 38.9%, *p* = 0.0404, Table 3). Pain improvement was significantly associated with the Karnofsky performance status scale (*p* = 0.0385), EORTC QLQ-C30 physical (*p* = 0.0115) and cognitive (*p* = 0.0078) functioning, 1st line chemotherapy regimen (*p* = 0.0036), and response to chemotherapy (*p* = 0.0348). 

Mean pain scores decreased significantly compared with baseline in the EPC group, with estimated mean ± SE scores of 4.83 ± 0.11 at baseline, 3.11 ± 0.16 at 1 month, 2.90 ± 0.22 at 3 months, 3.46 ± 0.29 at 6 months, and 2.82 ± 0.40 at 12 months (*p* < 0.0001) (Figure 2A). Although mean pain scores differed significantly between time points (*p* < 0.0001), there was no statistically significant difference between the two groups (*p* = 0.2157). Mean depression scores did not differ significantly in the two groups (*p* = 0.7792) or between time points (*p* = 0.1488) (Figure 2B). However, at 12 months, the mean depression score was lower in the EPC than in the control group.

Patients in both study groups reported improvements in assessment of pain intensity by week 4. Mean decrease in score was significantly greater in the EPC than in the usual care group (1.5-point vs. 1.0-points, *p* = 0.0318, Figure 3, Appendix A).

### 3.3. Health-Related Quality of Life and Cointerventions

At 4 weeks, patients in the EPC group reported significant increases from baseline in EORTC QLQ-C30 general questionnaire scores of global health status, role of functioning, nausea and vomiting, and pain, whereas patients in the usual care group reported significant reductions in scores of physical functioning, role of functioning, fatigue, pain, and dyspnea. The scores of the role of functioning and fatigue differed significantly in the EPC and UOC groups (Figure 3, Appendix A). 

Although, at 4 weeks, sleep disturbance score did not differ significantly between the EPC and UOC groups, patients in the EPC group reported a significant decrease from baseline in the sleep disturbance score (Table 4). Satisfaction with pain management, patient’s and investigator’s global assessment, and CGI-I scores differed significantly between the two groups 4 weeks after enrollment (Table 5).

The proportions of patients in the EPC and UOC groups with celiac plexus neurolysis (CPN; 16% vs. 18%) and the proportions that consulted a psychiatrist (12% vs. 12%) were similar in the two groups. By April 2017, 243 (84.4%) patients had died. There was no difference in overall survival between the two groups (Figure 3).

## 4. Discussion

In this study of 288 patients with advanced pancreatobiliary cancer, EPC did not significantly increase the proportion of patients with ≥50% reductions in pain and depression scores from baseline to week 4. However, the proportion of patients with BPI worst pain score ≤3, which may be a surrogate for a clinically meaningful endpoint, was significantly higher in the EPC group than in the usual care group (51.1% vs. 38.9%, *p* = 0.0404). 

The primary outcome of the present study may be considered unusual, as the pain scores of 243 (84.4%) patients at baseline were between 4 and 6 (Appendix A). Our results indicate that severe pain and symptoms of depression are not as prevalent in patients newly diagnosed with advanced pancreatobiliary cancer as generally believed. However, scores on global health status, role of functioning, nausea and vomiting, pain in the EORTC QLQ-C30 general questionnaire, satisfaction with pain management, patient’s and investigator’s global assessment, and CGI-I were significantly better in the intervention than in the control group 4 weeks after enrollment. When we used an anchor-based approach for the estimates of the minimal clinically important difference for EORTC QLQ-C30 in the EPC group [20], changes of 20.61 (95% CI: 14.88, 26.34) units on the role, 19.45 (15.4, 23.5) on the social functioning scale, respectively; −15.15 (−21.62, −8.68) on the insomnia scale may be clinically meaningful (Appendix A). Although the between-group differences in the proportion of patients with ≥50% reductions in pain and depression score from baseline to week 4, as determined by the BPI worst pain score and the CES-D, were not statistically significant, this trial shows promising findings that support a selective approach to early palliative care for patients with advanced PC and BTC.

Specialist palliative care and early palliative care may have slight effects on QoL in cancer patients, including those with pancreatic cancer [12,14,21]. A meta-analysis assessing the effects of palliative care on pain and other secondary outcomes yielded inconclusive results [12]. Previous trials did not include integration of specialist palliative care triggered by patients’ needs. All patients newly diagnosed with PC and BTC should undergo a full assessment of symptom burden, psychological status, and social support as early as possible. The present study enrolled patients with cancer-related pain (BPI worst pain score >3) and/or depression (CES-D > 16) within 8 weeks of diagnosis. As the mechanisms inducing pain in patients with PC and chronic pancreatitis are similar, assessment tools used for the latter may be applied to the former [22]. Numerical scales are frequently utilized to assess the intensity of pain, but the temporal nature of pain may be a more important determinant of health-related QoL than pain severity [23]. EPC may be more effective by screening those patients with unmet needs. 

The percentage of patients with a depressed mood was higher in patients with advanced PC and BTC (77/376, 20%) than in other cancers [24], with many patients reporting disturbed sleep [25]. Our EPC group reported a significant decrease in sleep disturbance score from baseline. Compared with usual care, a supportive–expressive psychotherapeutic intervention was recently reported to help to relieve and prevent depressive symptoms in patients with advanced diseases, as well as helping these patients to address preparations for the end of life [26]. Further research is needed to explore the feasibility and effectiveness of practical psychotherapeutic interventions in patients with advanced PC and BTC.

Our EPC intervention involved two physician experts in PC and BTC rather than palliative care specialists. These two physicians were involved in both oncologic care and endoscopic interventions, providing advantages in the management of biliary and duodenal obstructions. Early application of advanced endoscopic procedures, such as endoscopic retrograde cholangiopancreatography (ERCP), endoscopic ultrasound (EUS), and enteral stenting, can have a significant impact on oncologic care through the rapid improvement of biliary and duodenal obstruction [27].

In a recent study [4], all patients with non-small cell lung cancer stages IIIb and IV were referred to the intervention, even if they did not have major symptoms or other distressing conditions, such as spiritual or social problems. However, many palliative care specialists question the feasibility, practicability, and efficiency of this approach [13]. Because of resource allocation issues, it may not be feasible to refer all patients at the palliative stage of their disease to specialist teams. All previous trials provided specialist palliative care for all patients, even if they did not have major symptoms, and neglected the potential role of general palliative care. Although physicians refer patients with complex needs to specialist palliative care, they must be capable and willing to deal with basic needs for palliative care.

Early referral for CPN has been reported to reduce pain and may reduce opioid consumption, resulting in fewer opiate-related complications, such as constipation in patients with painful, advanced PC [28,29]. However, the effects of CPN on QoL and survival remain unclear [30], and several studies have reported serious and fatal complications of CPN [31]. In the present study, only 16% of patients received CPN during the follow-up period.

The effect of EPC may depend on the primary tumor [5]. Compared with other malignancies, advanced PC and BTC cause significant morbidities, even in patients treated with current anticancer therapies. Almost all patients with locally advanced or metastatic PC experience cancer-related pain [8]. Palliative care has been shown to ameliorate the symptoms of pancreatic cancer, as well as those from its treatment [32]. However, evidence for the effects of late palliative care is ambiguous because the time required to establish its beneficial effects may be too short [12]. Palliative care applied early, around the time of diagnosis of advanced or metastatic cancer, may better improve symptoms and the adverse effects of anticancer therapy. According to the latest consensus definition, palliative care is regarded as early when administered within 8 weeks of diagnosis of advanced cancer [33]. In the present study, patients were enrolled within 8 weeks after diagnosis.

Although assessment rates were similar in both groups and uniformly high, only around 10% of patients could be assessed at 12 months, mainly because of their high mortality rates. The EPC team consisted of nurses and attending physicians. Some patients in the control group also received palliative care, but this was limited to advice on the telephone. Lack of blinding is always an issue in clinical research regarding early palliative care for patients with advanced cancer. This bias could not be avoided due to the lack of patient blinding in this prospective, randomized controlled trial. Other limitations of this trial include the accessibility of a well-functioning EPC. The exposure of control subjects to the intervention was unavoidable because the same attending physicians and nursing staff were involved in the care of both groups. This may have led to an underestimation of the true effects of the active intervention.

## 5. Conclusions

Although the primary outcome was not met, this trial showed promising findings that EPC may improve early pain relief for patients with advanced pancreatobiliary cancers, compared to usual oncologic care. An ongoing randomized controlled trial is testing the hypothesis that EPC may benefit patients with metastatic upper gastrointestinal cancers, including PC and BTC, treated with first-line chemotherapy [34]. Further research may help to design a strategy to implement EPC compatible with current health service provisions for patients with pancreatobiliary cancers.

## Figures and Tables

**Figure 1 cancers-11-00079-f001:**
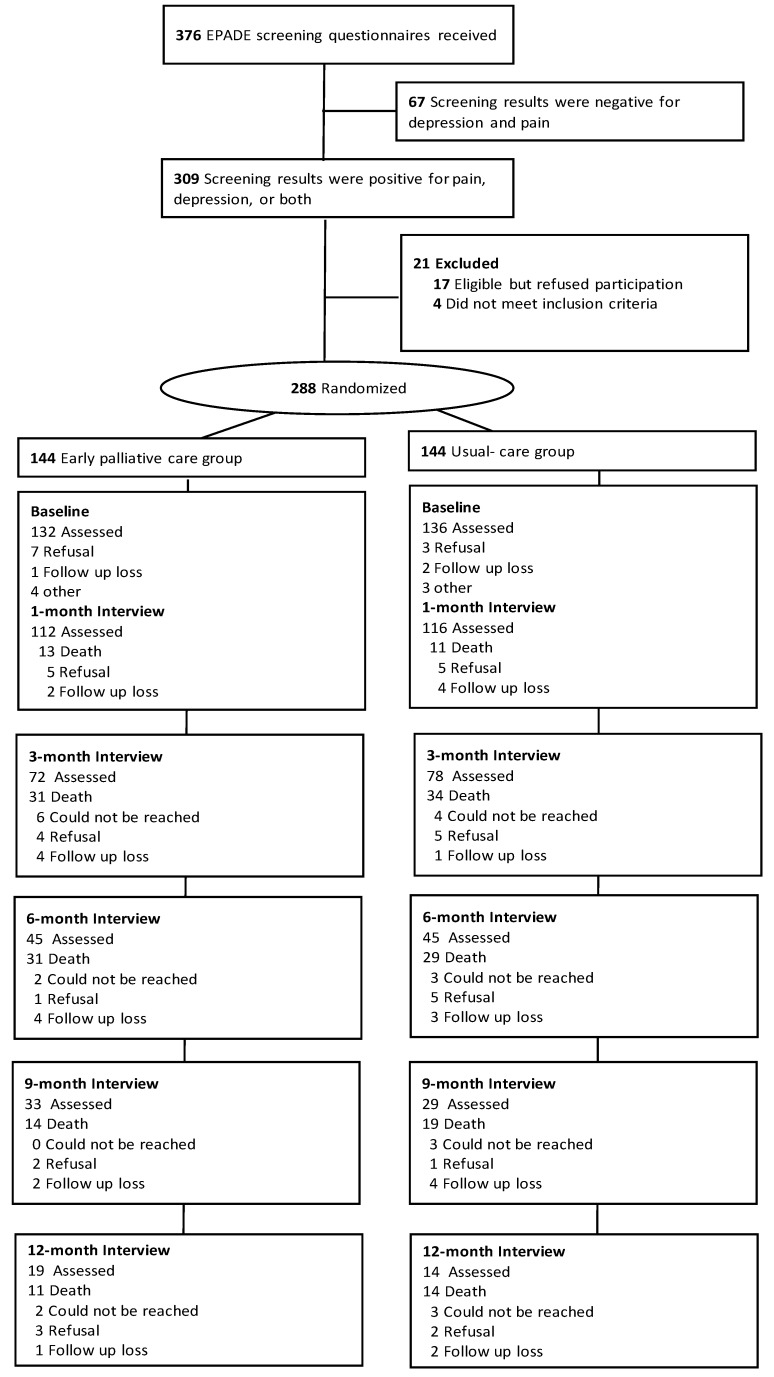
Trial profile. EPADE: Effect of Early Management on PAin and DEpression in Patients With PancreatoBiliary Cancer

**Figure 2 cancers-11-00079-f002:**
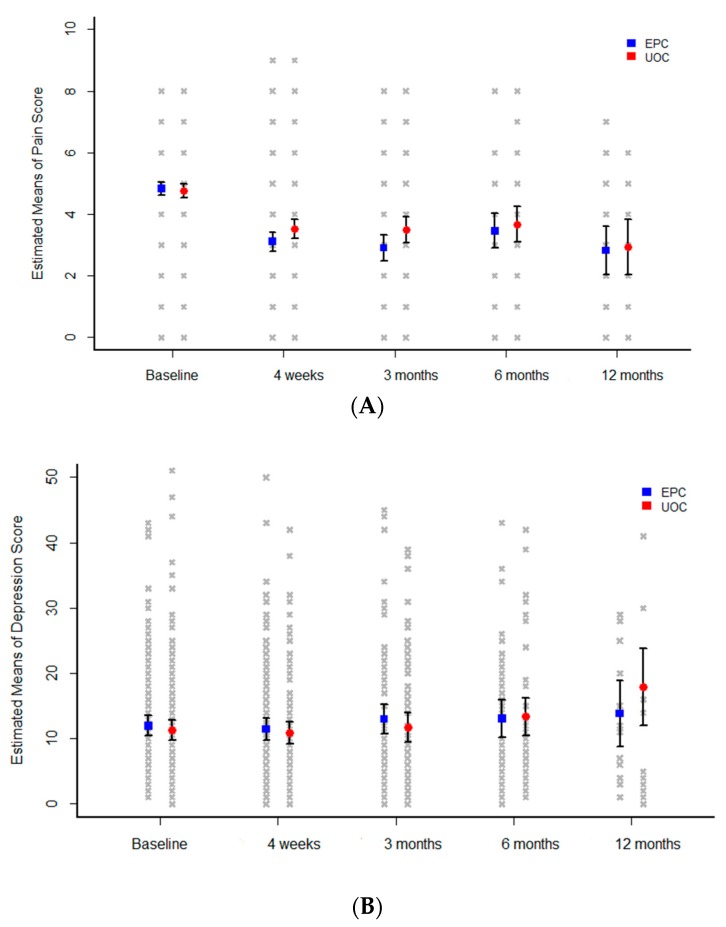
Mean changes in (**A**) pain scores and (**B**) depression scores among patients in the EPC (blue) and UOC (red) groups. EPC: Early palliative care, UOC: Usual oncology care.

**Figure 3 cancers-11-00079-f003:**
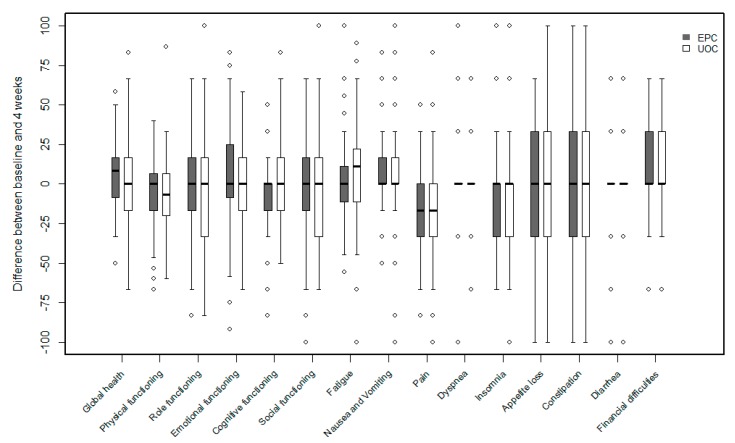
Box plot of differences of quality of life between baseline and 4 weeks.

**Table 1 cancers-11-00079-t001:** Baseline characteristics.

Characteristics	EPC (*n* = 144)	UOC (*n* = 144)	*p*-Value
Age, year	66.00 (40.00–86.00)	67.00 (42.00–89.00)	0.6954
Male	64 (44.44)	65 (45.14)	0.9057
Primary tumor site			0.9738
Pancreatic cancer	110 (76.39)	109 (6.22)	
Biliary cancer	34 (23.61)	34 (23.78)	
Metastasis			0.8396
No	17 (11.81)	18 (12.59)	
Yes	127 (88.19)	125 (87.41)	
Symptom type			0.9902
Pain only	105 (72.92)	106 (73.61)	
Depression only	5 (3.47)	5 (3.47)	
Pain and depression	34 (23.61)	33 (22.92)	
Kamofsky performance rating scales	70.00 (50.00–90.00)	70.00 (50.00–100.00)	0.8566
Pain score	4.00 (0.00–9.00)	4.00 (0.00–8.00)	0.8035
Depression score	9.00 (1.00–43.00)	8.00 (0.00–51.00)	0.2089

Results reported as median (range) or number (%). EPC: Early palliative care, UOC: Usual oncology care.

**Table 2 cancers-11-00079-t002:** Primary Outcome.

Variables	ITT Set (N = 288)			Modified ITT Set (N = 277)		
	EPC	UOC	*p*-Value	EPC	UOC	*p*-Value
Pain			0.4194			0.4616
<50%	98 (70.50)	104 (74.82)		94 (69.63)	98 (73.68)	
≥50%	41 (29.50)	35 (25.18)		41 (30.37)	35 (26.32)	
Depression			0.5732			0.5691
<50%	27 (69.23)	24 (63.16)		26 (68.42)	23 (62.16)	
≥50%	12 (30.77)	14 (36.84)		12 (31.58)	14 (37.84)	

Results reported as number (%). ITT: Intention to treat, EPC: Early palliative care, UOC: Usual oncology care.

**Table 3 cancers-11-00079-t003:** Proportion of patients in the EPC and UOC groups with brief pain inventory (BPI) worst pain score ≤3 or Center for Epidemiological Studies—Depression Scale (CES-D) score ≤15 at 4 weeks after enrollment.

Variables	ITT set (*n* = 288)			Modified ITT set (*n* = 277)		
	EPC	UOC	*p*-Value	EPC	UOC	*p*-Value
Pain			0.0404			0.0364
<3 score	68 (48.92)	85 (61.15)		64 (47.41)	80 (60.15)	
≥3 score	71 (51.08)	54 (38.85)		71 (52.59)	53 (39.85)	
Depression			0.9234			0.9206
<15 score	23 (58.97)	22 (57.89)		22 (57.89)	21 (56.76)	
≥15 score	16 (41.03)	16 (42.11)		16 (42.11)	16 (43.24)	

Results reported as number (%).

**Table 4 cancers-11-00079-t004:** Pain intensity and sleep disturbance, for all patients 4 weeks after enrollment.

Variables	Baseline	1 Month	*p*-Value ^†^	Diff	*p*-Value ^‡^
Assessment of pain intensity					0.0318
EPC	4.00 (0.00–9.00)	3.00 (0.00–8.00)	<0.0001	−1.50 (−6.00–4.00)	
UOC	4.00 (0.00–8.00)	4.00 (0.00–9.00)	<0.0001	−1.00 (−6.00–3.00)	
Assessment of sleep disturbance					0.3157
EPC	7.50 (0.00–23.00)	5.00 (0.00–27.00)	0.0112	−2.00 (−19.00–26.00)	
UOC	6.00 (0.00–24.00)	5.00 (0.00–23.00)	0.1493	−1.00 (−16.00–18.00)	

Results reported as median (range), ^†^ Wilcoxon signed-rank test, ^‡^ Mann–Whitney U test.

**Table 5 cancers-11-00079-t005:** Satisfaction with pain management, patient’s and investigator’s global assessment, and clinical global impression–improvement scale (CGI-I), for all patients 4 weeks after enrollment.

Variables	EPC	UOC	*p*-Value
Satisfaction with pain control	2.00 (1.00–4.00)	2.00 (1.00–5.000)	0.0260
Investigator’s global assessment	3.00 (1.00–5.00)	3.00 (1.00–5.00)	0.0027
Patient’s global assessment	2.00 (1.00–5.00)	3.00 (1.00–5.00)	0.0005
CGI-I	3.00 (1.00–5.00)	3.00 (1.00–5.00)	0.0383

Results reported as median (range).

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
