# Peer review of "Effect of Early Management on Pain and Depression in Patients with Pancreatobiliary Cancer: A Randomized Clinical Trial"

_cancers, 2019, doi:10.3390/cancers11010079_

Reviewer 1 Report

The study deals with the use of early integrated palliative care for pain management in patients with pancreatobiliary cancer.

The topic is extremely important and interesting, particularly for the high mortality rate and distress of these patients. However, the article suffers from some problems.

 The methods section appears confused and with several repetitions. For instance, sample and study tools are mentioned in the study design paragraph and repeated later on. Paragraphs need to be better separated.

  created two wide categories (pain >3 or < 3 and depression >15 or<15). I wonder if it is realistic or if it would be more clinically useful to stratify according to the severity of pain/depression. Moreover, outcomes are measured at 12 months and this leads to confusion. Several patients died early: were they in worse clinical conditions?

Among the secondary outcomes, the pain intensity appears. Which is the difference between this outcome and the primary one?

In the section 2.2 “Patients”, the authors did not specify whether the patients were outpatient or inpatient. Lines from 100 to 103 repeat the paragraph 2.1. This should be avoided to facilitate reading.

In the section 2.3 “Intervention”, a more detailed  explanation of what the intervention consists of is needed. Is it led by nurses? How often? Is the intervention different from the usual physician’s visits? What did the psychiatrist do in the two groups? And what about the controls? How long did it last? Who made telephone follow-up and who did the visits? More precision in describing the intervention is required, the current shape is not acceptable.

In the section 2.5 “Measures”, authors should report all the tools adopted and better explain which instrument was used for each outcome; moreover, authors should specify if tools were validated or not.

Figure 1 is not clear enough. Changes are needed to clarify numbers. Why were 21 patients excluded?

Table 1 is badly laid out. I do not agree in considering pain and depression as dichotomous variables. For the patients, having a pain intensity of 5 is extremely different from having a pain intensity of 9. I suggest authors to redo the statistical analyses according to clinical severity. I think that current statistical analyses may have flattened the results.

Tables  are not readable-friendly (e.g., table 4 goes on for 3 pages). Tables should be summarized reporting only the most significant data.

The discussion needs to be rewrited according to the results. 

Author Response

Dear Editors-in-Chief, Cancers

We thank the Reviewer for his/her thoughtful and expert review and suggestion of our manuscript “Effect of Early Management on Pain and Depression in Patients with Pancreatobiliary Cancer: A Randomized Clinical Trial”. and for valuable and insightful comments. Our responses to the Reviewer’s comments are as follows:

Point-to-Point Responses to Comments

Reviewer #1 (Comments to the Author (Required)): 

The study deals with the use of early integrated palliative care for pain management in patients with pancreatobiliary cancer. The topic is extremely important and interesting, particularly for the high mortality rate and distress of these patients. However, the article suffers from some problems.

The methods section appears confused and with several repetitions. For instance, sample and study tools are mentioned in the study design paragraph and repeated later on. Paragraphs need to be better separated.

Response: In response of the reviewer’s comment, we removed “ The co-primary endpoints were defined as 50% reductions from baseline in BPI pain score and CES-D score after 4 weeks” and separated paragraph in the 2.6 Statistical analysis.

 The authors set a reduction of 50% in the pain and depression scores at 4 weeks as endpoints but then created two wide categories (pain >3 or < 3 and depression >15 or<15). I wonder if it is realistic or if it would be more clinically useful to stratify according to the severity of pain/depression. Moreover, outcomes are measured at 12 months and this leads to confusion. Several patients died early: were they in worse clinical conditions?

Response: In clinical research and in many clinical settings, pain intensity is commonly measured with numerical rating scales (NRS), where patients rate their pain intensity from 0 to 10 or 0 to 100. Although there is no clearly defined mechanism for classifying pain intensity into these categories. cancer guideline by the World Health Organization are often developed based upon categorical ratings of pain (e.g., 0-3, 4-6, 7-10, for mild, moderate and severe). Higher scores in the CES-D indicated greater depressive symptoms, with scores ≥16 indicating individuals at risk for clinical depression, with good sensitivity and specificity and high internal consistency.

Pancreatic cancer (PC) and biliary tract cancer (BTC) are one of the deadliest cancers. The prognosis of these cancer is very closely related with tumor biology and the response to the anticancer treatment. The survival time cannot be expected exactly by the clinical condition at the time of enrollment.

Among the secondary outcomes, the pain intensity appears. Which is the difference between this outcome and the primary one?

Response: In the present study, the primary endpoints were 50% reductions from baseline to week 4 in pain score and depression score, as determined by the BPI worst pain score and the CES-D, respectively. Pain intensity in the secondary outcomes means quantitative pain relief. In response of the reviewer’s comments, we changed the pain intensity to “ quantitative pain relief”.

In the section 2.2 “Patients”, the authors did not specify whether the patients were outpatient or inpatient. Lines from 100 to 103 repeat the paragraph 2.1. This should be avoided to facilitate reading.

Response: In response of the reviewers comments, we changed the sentence to “Patients who presented to the outpatient cancer clinic were invited by their physicians to enroll in the study:…”.

In the section 2.3 “Intervention”, a more detailed explanation of what the intervention consists of is needed. Is it led by nurses? How often? Is the intervention different from the usual physician’s visits? What did the psychiatrist do in the two groups? And what about the controls? How long did it last? Who made telephone follow-up and who did the visits? More precision in describing the intervention is required, the current shape is not acceptable.

Response: In the present study, the EPC intervention was led by research nurse team, trained in assessing symptom response and medication adherence; in providing specific education about pain and depression; and in making treatment adjustments according to NCCN guidelines. Follow-up intervention visits or telephone coaching were scheduled daily until BPI worst pain score was ≤3. After that, participants were contacted at baseline and at 1, 3, 6, 9, and 12 months. In addition, telephone calls were triggered when patients reported inadequate symptom improvement, nonadherence to medication, adverse effects, or suicidal ideation, or when patients requested to be contacted.

Patients randomized to the usual-care group received no formal intervention. Usual oncology care (UOC), provided to all patients, was directed by an attending physician and consisted of anticancer and symptom control treatments and consultation with psychiatric and pain care specialists. The latter were provided whenever requested, regardless of group assignment. The proportions that consulted a psychiatrist (12% vs 12%) were similar in the two groups.

In the section 2.5 “Measures”, authors should report all the tools adopted and better explain which instrument was used for each outcome; moreover, authors should specify if tools were validated or not.

Response: We adopted all the tools validated. We added 3 references for Satisfaction with pain management, Patient’s and Investigator’s Global Assessments, and CGI-I scale. Satisfaction with pain management was measured on a 5-point scale (1= very good; 5=very poor). [16. Thinh, D.H.Q.; Sriraj, W.; Mansor, M.; Tan, K.H.; Irawan, C..; Kurnianda, J.; Nguyen, Y.P.; Ong-Cornel, A.; Hadjiat, Y.; Moon, H. et al. Patient and Physician Satisfaction with Analgesic Treatment: Findings from the Analgesic Treatment for Cancer Pain in Southeast Asia (ACE) Study. Pain Res Manag. 2018, 2193710]. The 5 point Patient’s and Investigator’s Global Assessments were used as outcome measures with a score of 5 defined as very effective. [17. Nadstawek, J.; Leyendecker, P.; Hopp, M.; Ruckes, C.; Wirz, S.; Fleischer, W.; Reimer, K. Patient assessment of a novel therapeutic approach for the treatment of severe, chronic pain. Int J Clin Pract. 2008, 62, 1159–1167.]. Clinicians evaluated patients on the CGI-I scale, a 7 point scale measuring the improvement or worsening of a patient relative to baseline, with 1= very much improved; 5= very much worse. [18. Busner, J.; Targum, S.D. The clinical global impressions scale: applying a research tool in clinical practice. Psychiatry (Edgmont) 2007, 4, 28–37.]

Figure 1 is not clear enough. Changes are needed to clarify numbers. Why were 21 patients excluded?

Response: Before randomization, 17 eligible patients refused to take part in the present study and 4 did not meet the inclusion criteria. After randomization, we excluded 12 patients in EPC group (7 withdrawal of consent, 1 lost to follow-up, and 3 did not meet the inclusion criteria [pathology showed non-adenocarcinoma]) and 8 patients in UOC group [3 withdrawal of consent, 2 lost to follow-up, and 3 did not meet the inclusion criteria [pathology showed non-adenocarcinoma]). Thus 112 patients were assessed at 1-month and 20 not assessed (13 death, 5 refusal and 2 lost to follow-up) in EPC group. In UOC group, 116 patients were assessed at 1-month and 20 not assessed (11 death, 5 refusal and 4 lost to follow-up. We changed Figure 1 to include this data.

Table 1 is badly laid out. I do not agree in considering pain and depression as dichotomous variables. For the patients, having a pain intensity of 5 is extremely different from having a pain intensity of 9. I suggest authors to redo the statistical analyses according to clinical severity. I think that current statistical analyses may have flattened the results.

Response: In the present study, we enrolled patients with who had cancer-related pain (Brief Pain Inventory [BPI] worst pain score >3) and/or depression (Center for Epidemiological Studies-Depression Scale [CES-D] >16). The table showed the proportion of patients with BPI worst pain score > 3 and/or CES-D >16 that were the co-primary endpoints. As mentioned in the discussion session, the pain scores of 243 (84.4%) patients at baseline were between 4 and 6. We provide the distribution of BPI worst pain score and CES-D as supplementary Figure 1.

Tables are not readable-friendly (e.g., table 4 goes on for 3 pages). Tables should be summarized reporting only the most significant data.

Response: I agree with your comment. To describe readable-friendly, we changed the table 4 to figure 3 and the others were rearranged. We provided table 4 as Supplementary Table 2. 

Figure 3. Box plot of differences of quality of life between baseline and 4 weeks

The discussion needs to be rewritten according to the results.

Response: In response to the reviewer’s comments, we reported the minimal clinically important difference (MID) for EORTC QLQ-C30 questionnaire in the discussion session.

Reviewer 2 Report

Aim of this monocenter, randomized study was to assess if early palliative care (EPC), targeting pain and depression, integrated with usual oncologic care, could improve pain and depression in 288 patients with advanced pancreatobiliary cancers. The primary endpoints were ≥50% reductions from baseline to week 4 in pain and depression scores.

Authors find that EPC did not significantly increase the proportion of patients with ≥50% reductions in pain and depression scores from baseline to week 4 compared with usual oncologic care. The proportion of patients with Brief Pain Inventory worst pain score ≤3 was significantly higher (51% vs 39%) and the reduction in pain intensity score greater (1.5 vs 1.0 points) in the EPC group than in the usual care group.

Comments.

- Methods. The nature of intervention is important. Please report the make-up of palliative care team, the level of training of staff, and the setting of delivery (outpatient, inpatient, home care). Other than the timing of referral, did patients have differential treatment with palliative care (e.g. intensity of follow-up) in each arm?

- Methods. Missing data management should be stated.

- Results, Figure 1. 21 patients were excluded from the screening results. Please clarify because these patients were excluded. In the EPC group, 112 patients were assessed ad 1-month and 20 not assessed (13 death, 5 refusal and 2 lost to follow-up), thus 132 patients remain at 1-month. Since 144 patients were randomized in this group, why information on 12 patients were not available? And same goes for 8 patients in the usual-care group. Again, in Tables 2 and 3 were reported that pain reduction was evaluated on 139 patients in each group, different than reported in Figure 1.

- Results, Table 1. Pain score was presented as median value and range while depression score as mean value and SE. Why scores were different reported?

- Results, Table 4. How many patients were evaluated for Quality of life? Is it correct that median value of the global health status in the EPC group was 50 at baseline and 50 at 4 weeks (p=0.040), with a difference of 8.33? Why the authors did not report mean values?

- The proportion of patients who received palliative care in the usual-care group should be stated. Time from referral to death and mean number of visits for each group should also be reported.

- In the statistical analyses was reported that “A mixed model was used to analyze both the whole time-point and the group” but the result was not reported in the results section.

The reviewer think that this is the primary statistical analysis to perform.

- Please report the minimal clinically important difference (MID) for EORTC QLQ-C30 questionnaire. Please state if the magnitude of change was clinically meaningful.

Author Response

Dear Editors-in-Chief, Cancers

We thank the Reviewer for his/her thoughtful and expert review and suggestion of our manuscript “Effect of Early Management on Pain and Depression in Patients with Pancreatobiliary Cancer: A Randomized Clinical Trial”. and for valuable and insightful comments. Our responses to the Reviewer’s comments are as follows:

Reviewer 2

Aim of this monocenter, randomized study was to assess if early palliative care (EPC), targeting pain and depression, integrated with usual oncologic care, could improve pain and depression in 288 patients with advanced pancreatobiliary cancers. The primary endpoints were ≥50% reductions from baseline to week 4 in pain and depression scores.

Authors find that EPC did not significantly increase the proportion of patients with ≥50% reductions in pain and depression scores from baseline to week 4 compared with usual oncologic care. The proportion of patients with Brief Pain Inventory worst pain score ≤3 was significantly higher (51% vs 39%) and the reduction in pain intensity score greater (1.5 vs 1.0 points) in the EPC group than in the usual care group.

Comments.

- Methods. The nature of intervention is important. Please report the make-up of palliative care team, the level of training of staff, and the setting of delivery (outpatient, inpatient, home care). Other than the timing of referral, did patients have differential treatment with palliative care (e.g. intensity of follow-up) in each arm?

Response: In the present study, the EPC intervention was led by research nurse team, trained in assessing symptom response and medication adherence; in providing specific education about pain and depression; and in making treatment adjustments according to NCCN guidelines. We delivered the interventions by telephone or during regularly scheduled outpatient care. Follow-up intervention visits or telephone coaching were scheduled daily until BPI worst pain score was ≤3. After that, participants were contacted at baseline and at 1, 3, 6, 9, and 12 months.. In addition, telephone calls were triggered when patients reported inadequate symptom improvement, nonadherence to medication, adverse effects, or suicidal ideation, or when patients requested to be contacted.

Patients randomized to the usual-care group received no formal intervention. Usual oncology care (UOC), provided to all patients, was directed by an attending physician and consisted of anticancer and symptom control treatments and consultation with psychiatric and pain care specialists. The latter were provided whenever requested, regardless of group assignment. The proportions that consulted a psychiatrist (12% vs 12%) were similar in the two groups.

In response to the reviewer’s comment, we added the description “We delivered the interventions by telephone or during regularly scheduled outpatient care.” to the first paragraph of 2.3 Intervention.

- Methods. Missing data management should be stated.

Response: We believe that it would be appropriate to conduct a conservative analysis based on the ITT principles. Therefore, we considered the missing data only when analyzing the primary purpose. The missing value was imputed to the negative results because it was considered that the cause of the missing value was not effective for treatment. All other analyzes for secondary purposes excluded missing data. In response to the reviewer’s comment, we add the description to the 1st paragraph of 2.6 Statistical Analyses.

- Results, Figure 1. 21 patients were excluded from the screening results. Please clarify because these patients were excluded. In the EPC group, 112 patients were assessed ad 1-month and 20 not assessed (13 death, 5 refusal and 2 lost to follow-up), thus 132 patients remain at 1-month. Since 144 patients were randomized in this group, why information on 12 patients were not available? And same goes for 8 patients in the usual-care group. Again, in Tables 2 and 3 were reported that pain reduction was evaluated on 139 patients in each group, different than reported in Figure 1.

Response: Before randomization, 17 eligible patients refused to take part in the present study and 4 did not meet the inclusion criteria. After randomization, we excluded 12 patients in EPC group (7 withdrawal of consent, 1 lost to follow-up, and 3 did not meet the inclusion criteria [pathology showed non-adenocarcinoma]) and 8 patients in UOC group [3 withdrawal of consent, 2 lost to follow-up, and 3 did not meet the inclusion criteria [pathology showed non-adenocarcinoma]]. Thus 112 patients were assessed at 1-month and 20 not assessed (13 death, 5 refusal and 2 lost to follow-up) in EPC group. In UOC group, 116 patients were assessed at 1-month and 20 not assessed (11 death, 5 refusal and 4 lost to follow-up. We changed Figure 1 to include this data.

- Results, Table 1. Pain score was presented as median value and range while depression score as mean value and SE. Why scores were different reported?

Response: There was a typo. Both pain and depression scores were described as median and range because they did not satisfy the normality.

- Results, Table 4. How many patients were evaluated for Quality of life? Is it correct that median value of the global health status in the EPC group was 50 at baseline and 50 at 4 weeks (p=0.040), with a difference of 8.33? Why the authors did not report mean values?

Response: In the baseline, all patients responded to QoL questionnaire, and in the 4th week, all but 41 patients responded to the QoL questionnaire. The score of each attribute of the QoL questionnaire is distributed from 0 to 100 points. The nonparametric method is used to show the result conservatively because it does not satisfy the normality. It describes the median and range, which are descriptive statistics corresponding to this analysis method.

If the median values of the two groups are the same, the medians of the values for the differences are not the same. If you have any of the following types of data, the median for before and after will all be 50, but the median for difference will be l. The nonparametric method is an analytical technique that considers not only the median but also the sign and rank of each value, so the statically significance was obtained.

ID

Before

After

Diff

1

48

90

42

2

49

50

1

3

50

10

-40

4

51

70

19

5

52

30

-22

- The proportion of patients who received palliative care in the usual-care group should be stated. Time from referral to death and mean number of visits for each group should also be reported.

Response: Usual oncology care (UOC), provided to all patients in the usual-care group, was directed by an attending physician and consisted of anticancer and symptom control treatments and consultation with psychiatric and pain care specialists. As mention in the results, The proportions of patients in the EPC and UOC groups with celiac plexus neurolysis (CPN; 16% vs. 18%) and the proportions that consulted a psychiatrist (12% vs 12%) were similar in the two groups. They were not referred to palliative care specialists.

- In the statistical analyses was reported that “A mixed model was used to analyze both the whole time-point and the group” but the result was not reported in the results section.

Response: In response to the reviewer’s comment, we removed the sentence.

The reviewer think that this is the primary statistical analysis to perform.

- Please report the minimal clinically important difference (MID) for EORTC QLQ-C30 questionnaire. Please state if the magnitude of change was clinically meaningful.

Response: As the anchor-based approach has received much attention in the literature [Musoro ZJ et al 2018], we used an anchor-based approach for the estimates of the minimal clinically important difference for EORTC QLQ-C30 in EPC group [Jaeschke, R.; Singer, J.; Guyatt, G.H. Measurement of health status. ascertaining the minimal clinically important difference. Control Clin Trials. 1989, 10, 407-415].

The magnitude of change in global health status, physical functioning, role functioning, emotional functioning, social functioning, fatigue, pain, insomnia, appetite loss, and financial difficulty were statistically significant. However, the literature does not clearly provide an evidence-based approach to determine MIDs for scores of the EORTC QLQ-C30 [Musoro ZJ et al 2018]. Changes of 20.61 (95% CI: 14.88, 26.34) units on the role, 19.45 (15.4, 23.5) on the social functioning scale, respectively; -15.15(-21.62, -8.68) on the insomnia scale may be clinically meaningful (Appendix Supplementary Table 3). We added the description to 2nd paragraph of the discussion session.

Supplementary Table 3. Mean (SD) of QLQ-C30 change scores in the three anchor-defined groups and the difference in mean change scores (95% CI) between adjacent categories. The number of patients in the anchor-defined groups varies by the scale and is therefore presented as a range of values for all the scales. Difference in mean change refers to the difference in mean of QLQ-C30 change scores between the “improvement” and “no change” (improvement) and between the “no change” and “deterioration” (deterioration)

QLQ-C30

Improvement

No   change

Deteriorated

minimal clinically   important difference, 95% CI

Improvement

Deteriorated

Global health   status

13.67±18.14

5.30±19.34

-4.84±23.55

8.37(4.84,   11.9)*

-10.14(-15.11,   -5.17)*

Physical   functioning

5.87±18.49

-3.88±22.65

-13.76±27.1

9.75(5.85,   13.65)*

-9.88(-15.63,   -4.13)*

Role functioning

20.00±31.55

-0.61±29.39

-8.06±33.57

20.61(14.88,   26.34)*

-7.45(-14.67,   -0.23)*

Emotional   functioning

17.67±26.06

3.33±24.35

-8.60±30.69

14.34(9.6,   19.08)*

-11.93(-18.35,   -5.51)*

Cognitive   functioning

-4.00±23.21

-2.73±21.22

-4.84±28.61

-1.27(-5.45,   2.91)

-2.11(-7.99,   3.77)

Social functioning

17.33±21.77

-2.12±21.29

-25.81±26.82

19.45(15.4, 23.5)*

-23.69(-29.3,   -18.08)*

Fatigue

-11.56±23.67

4.24±26.92

6.45±35.81

-15.8(-20.58,   -11.02)*

2.21(-5.18,   9.6)

Nausea and Vomiting

2.67±22.40

12.12±28.59

4.30±23.16

-9.45(-14.3,   -4.6)*

-7.82(-13.49,   -2.15)*

Pain

-20.00±27.22

-14.55±28.34

-13.98±29.22

-5.45(-10.68,   -0.22)*

0.57(-5.92,   7.06)

Dyspnea

5.33±28.35

4.85±24.36

4.30±35.22

0.48(-4.48,   5.44)

-0.55(-7.68,   6.58)

Insomnia

-20.00±31.91

-4.85±36.53

-7.53±41.01

-15.15(-21.62,   -8.68)*

-2.68(-11.54,   6.18)

Appetite loss

-16.00±43.16

-5.45±47.04

2.15±42.11

-10.55(-19.06,   -2.04)*

7.6(-2.27,   17.47)

Constipation

2.67±40.73

0.00±35.14

-2.15±38.43

2.67(-4.48,   9.82)

-2.15(-10.52,   6.22)

Diarrhea

-1.33±17.95

2.42±26.34

-1.08±31.6

-3.75(-8.01,   0.51)

-3.5(-10.2,   3.2)*

Financial   difficulties

-37.33±11.06

0.00±0.00

35.48±8.32

-37.33(-38.78,   -35.88)*

35.48(33.99,   36.97)*

*Differences that are statistically significant

Round  2

Reviewer 2 Report

The authors responded to questions.